# Variational Hyper RNN for Sequence Modeling

## Abstract

In this work, we propose a novel probabilistic sequence model that excels at capturing high variability in time series data, both across sequences and within an individual sequence. Our method uses temporal latent variables to capture information about the underlying data pattern and dynamically decodes the latent information into modifications of weights of the base decoder and recurrent model. The efficacy of the proposed method is demonstrated on a range of synthetic and real-world sequential data that exhibit large scale variations, regime shifts, and complex dynamics.

## 1 Introduction

Recurrent neural networks (RNNs) are the natural architecture for sequential data as they can handle variable-length input and output sequences. Initially invented for natural language processing, long short-term memory (LSTM; Hochreiter & Schmidhuber 1997), gated recurrent unit (GRU; Cho et al. 2014) as well as the later attention-augmented versions (Vaswani et al., 2017) have found wide-spread successes from language modeling (Mikolov et al., 2010; Kiros et al., 2015; Jozefowicz et al., 2016) and machine translation (Bahdanau et al., 2014) to speech recognition (Graves et al., 2013) and recommendation systems (Wu et al., 2017). However, RNNs use deterministic hidden states to process input sequences and model the system dynamics using a set of time-invariant weights, and they do not necessarily have the right inductive bias for time series data outside the originally intended domains.

Many natural systems have complex feedback mechanisms and numerous exogenous sources of variabilities. Observations from such systems would contain large variations both across sequences in a dataset as well as within any single sequence; the dynamics could be switching regimes drastically, and the noise process could also be heteroskedastic. To capture all these intricate patterns in RNN with deterministic hidden states and a fixed set of weights requires learning about the patterns, the subtle deviations from the patterns, the conditions under which regime transitions occur which is not always predictable. Outside of the deep learning literature, many time series models have been proposed to capture specific types of high variabilities. For instance, switching linear dynamical models (Ackerson & Fu, 1970; Ghahramani & Hinton, 1996; Murphy, 1998; Fox et al., 2009) aim to model complex dynamical systems with a set of simpler linear patterns. Conditional volatility models (Engle, 1982; Bollerslev, 1986) are introduced to model time series with heteroscedastic noise process whose noise level itself is a part of the dynamics. However, these models usually encode specific inductive biases in a hard way, and cannot learn different behaviors and interpolate among the learned behaviors as deep neural nets.

In this work, we propose a new class of neural recurrent latent variable model, called the *variational hyper RNN* (VHRNN), which can perform dynamic regime identification and re-identification dynamically at inference time. Our model captures complex time series without encoding a large number of patterns in static weights, but instead only encodes base dynamics that can be selected and adapted based on run time observations. Thus it can easily learn to express a rich set of behaviors including but not limited to the ones mentioned above. Our model can dynamically identify the underlying pattern, express uncertainty due to observation noise, lack of information, or model misspecification. As such, VHRNN can model complex patterns with fewer parameters; and when given lots of parameters, it generalizes better than previous methods.

The VHRNN is built upon the previous variational RNN (VRNN) models (Chung et al., 2015) and hypernetworks (Ha et al., 2016). The VRNN models introduce stochastic latent variables at every time step, which are inferred using a variational recognition model. The overall model is trained by maximizing the evidence lower bound (ELBO). In VRNN, the latent variables capture the information in the stochastic hidden states and are then fed as input to the RNN and decoding model to produce reconstructed observations. While in our work, the latent variables are decoded to produce the RNN transition weights and observation projection weights in the style of hypernetworks (Ha et al., 2016), i.e., dynamically generating the scaling and bias vectors to adjust the base weights of the RNN. We demonstrate that the proposed VHRNN model is better at capturing different types of variability on several synthetic as well as real-world time series datasets.

## 2 BACKGROUND AND RELATED WORK

**Variational Autoencoder** Variational autoencoder (VAE) is one of the most popular unsupervised approaches to learning a compact representation from data (Kingma & Welling, 2013). It uses a variational distribution $q(\mathbf{z}|\mathbf{x})$ to approximate the intractable posterior distribution of the latent variable $\mathbf{z}$. With the use of variational approximation, it maximizes the evidence lower bound (ELBO) of the marginal log-likelihood of data

$$\mathcal{L}(\mathbf{x}) = \mathbb{E}_{q(\mathbf{z}|\mathbf{x})}[\log p(\mathbf{x}|\mathbf{z})] - D_{\mathrm{KL}}(q(\mathbf{z}|\mathbf{x}) \parallel p(\mathbf{z})) \leq \log p(\mathbf{x}),$$

where $p(\mathbf{z})$ is a prior distribution of $\mathbf{z}$ and $D_{\mathrm{KL}}$ denotes the Kullback–Leibler (KL) divergence. The approximate posterior $q(\mathbf{z}|\mathbf{x})$ is usually formulated as a Gaussian with a diagonal covariance matrix.

**Variational RNN for Sequential Data** Variational autoencoders have demonstrated impressive performance on non-sequential data like images. Many following works (Bowman et al., 2015; Chung et al., 2015; Fraccaro et al., 2016; Luo et al., 2018) extend the domain of VAE models to sequential data. Among them, variational RNN (VRNN; Chung et al. 2015) further incorporate a latent variable at each time step into their models. A prior distribution conditioned on the contextual information and a variational posterior is proposed at each time step to optimize a step-wise variational lower bound. Sampled latent variables from the variational posterior are decoded into the observation at the current time step. The VHRNN model makes use of the same factorization of sequential data and joint distribution of latent variables as in VRNN. However, in VHRNN model, the latent variables also parameterize the weights for decoding and transition in RNN cell across time steps, giving the model more flexibility to deal with variations within and across sequences.

**Importance Weighted Autoencoder and Filtering Variational Objective** A parallel stream of work to improve latent variable models with variational inference study tighter bounds of the data's log-probability than ELBO. Importance Weighted Autoencoder (IWAE; Burda et al. 2016) estimates a different variational bound of the log-likelihood, which is provably tighter than ELBO. Filtering Variational Objective (FIVO; Maddison et al. 2017) exploits the temporal structure of sequential data and uses particle filtering to estimate the data log-likelihood. FIVO still computes a step-wise IWAE bound based on the sampled particles at each time step, but it shows better sampling efficiency and tightness than IWAE. We use FIVO as the objective to train and evaluate our models.

**HyperNetworks** Our model is motivated by HyperNetworks (Ha et al., 2016) which use one network to generate the parameters of another. The dynamic version of HyperNetworks can be applied to sequence data, but due to lack of latent variables, can only capture uncertainty in the output variables. For discrete sequence data such as text, categorical output variables can model multi-model outputs very well; but on continuous time series with the typical Gaussian output variables, the model is much less capable at dealing with stochasticity. Furthermore, it does not allow straightforward interpretation of the model behaviour using the time-series of KL divergence as we do in Sec. 4. With the augmentation of latent variables, VHRNN is much more capable of modelling uncertainty. It is worth noting that Bayesian HyperNetworks (Krueger et al., 2017) also have a latent variable in the context of Hypernetworks. However, the goal of Bayesian HypernNtwork is an improved version of Bayesian neural net to capture model uncertainty. The work of Krueger et al. (2017) has no recurrent structure and cannot be applied to sequential data. Furthermore, the use of normalizing flow dramatically limits the flexibility of the decoder architecture design, unlike in VHRNN. Dezfouli et al. (2019) recently proposed to learn a disentangled low-dimensional latent space such that samples from the latent space are used to generate parameters of an RNN model. Different latent variables could account for the behaviours of different subjects. In spite of similarity in combining RNN

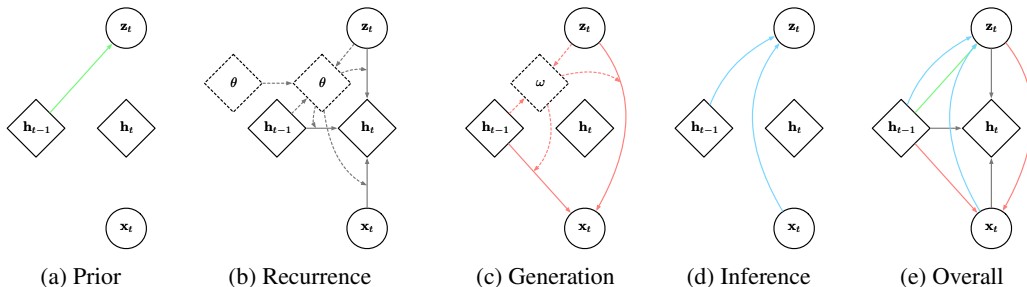

(a) Prior  (b) Recurrence  (c) Generation  (d) Inference  (e) Overall

Figure 1: Diagrams of the variational hyper RNN. Operators are indicated by arrows in different colors, and dashed lines and boxes represent the hypernetwork components. (a) Prior distribution in Eq. 3. (b) Recurrent model in Eq. 1. (c) Generative model in Eq. 2. (d) Inference model in Eq. 5. (e) The overall computational path. The hypernetwork components are left out.

with HyperNetworks, the motivations and architectures are fundamentally different. The work of Dezfouli et al. (2019) intends to learn a low-dimensional interpretable latent space that represents the difference between subjects in decision-making process while our models tries to better handle the variances both within and across sequences in general time-series data modeling. The work of Dezfouli et al. (2019) generate a set of parameters that is shared across time steps for each latent variable. In contrast, our model samples a latent variable and dynamically generates non-shared weights at each time step, which we believe is essential for handling variance of dynamics within sequences.

## 3 MODEL FORMULATION

**Variational Hyper RNN** A recurrent neural network (RNN) can be characterized by $\mathbf{h}_t = g_\theta(\mathbf{x}_t, \mathbf{h}_{t-1})$, where $\mathbf{x}_t$ and $\mathbf{h}_t$ are the observation and hidden state of the RNN at time step $t$, and $\theta$ is the fixed weights of the RNN model. The hidden state $\mathbf{h}_t$ is often used to generate the output for other learning tasks, e.g., predicting the observation at the next time step. We augment the RNN with a latent random variable $\mathbf{z}_t$, which is also used to output the non-shared parameters of the RNN at time step $t$.

$$\mathbf{h}_t = g_{\theta(\mathbf{z}_t, \mathbf{h}_{t-1})}(\mathbf{x}_t, \mathbf{z}_t, \mathbf{h}_{t-1}), \tag{1}$$

where $\theta(\mathbf{z}_t, \mathbf{h}_{t-1})$ is a hypernetwork that generates the parameters of the RNN at time step $t$. The latent variable $\mathbf{z}_t$ can also be used to determine the parameters of the generative model $p(\mathbf{x}_t | \mathbf{z}_{\leq t}, \mathbf{x}_{<t})$:

$$\mathbf{x}_t | \mathbf{z}_{\leq t}, \mathbf{x}_{<t} \sim \mathcal{N}(\boldsymbol{\mu}_t^{\text{dec}}, \boldsymbol{\Sigma}_t^{\text{dec}}), \quad \text{where } (\boldsymbol{\mu}_t^{\text{dec}}, \boldsymbol{\Sigma}_t^{\text{dec}}) = \phi_{\omega(\mathbf{z}_t, \mathbf{h}_{t-1})}^{\text{dec}}(\mathbf{z}_t, \mathbf{h}_{t-1}). \tag{2}$$

We hypothesize that the previous observations and latent variables, characterized by $\mathbf{h}_{t-1}$, define a prior distribution $p(\mathbf{z}_t | \mathbf{x}_{<t}, \mathbf{z}_{<t})$ over the latent variable $\mathbf{z}_t$,

$$\mathbf{z}_t | \mathbf{x}_{<t}, \mathbf{z}_{<t} \sim \mathcal{N}(\boldsymbol{\mu}_t^{\text{prior}}, \boldsymbol{\Sigma}_t^{\text{prior}}), \quad \text{where } (\boldsymbol{\mu}_t^{\text{prior}}, \boldsymbol{\Sigma}_t^{\text{prior}}) = \phi^{\text{prior}}(\mathbf{h}_{t-1}). \tag{3}$$

Eq. 2 and 3 result in the following generation process of sequential data:

$$p(\mathbf{x}_{\leq T}, \mathbf{z}_{\leq T}) = \prod_{t=1}^{T} p(\mathbf{z}_t | \mathbf{x}_{<t}, \mathbf{z}_{<t}) p(\mathbf{x}_t | \mathbf{x}_{<t}, \mathbf{z}_{\leq t}). \tag{4}$$

The true posterior distributions of $\mathbf{z}_t$ conditioned on observations $\mathbf{x}_{\leq t}$ and latent variables $\mathbf{z}_{<t}$ are intractable, posing a challenge in both sampling and learning. Therefore, we introduce an approximate posterior $q(\mathbf{z}_t | \mathbf{x}_{\leq t}, \mathbf{z}_{<t})$ such that

$$\mathbf{z}_t | \mathbf{x}_{\leq t}, \mathbf{z}_{<t} \sim \mathcal{N}(\boldsymbol{\mu}_t^{\text{enc}}, \boldsymbol{\Sigma}_t^{\text{enc}}), \quad \text{where } (\boldsymbol{\mu}_t^{\text{enc}}, \boldsymbol{\Sigma}_t^{\text{enc}}) = \phi^{\text{enc}}(\mathbf{x}_t, \mathbf{h}_{t-1}). \tag{5}$$

This approximate posterior distribution enables the model to be trained by maximizing a variational lower bound, e.g., ELBO (Kingma & Welling, 2013), IWAE (Burda et al., 2016) and FIVO (Maddison et al., 2017). We refer to the main components of our model, including $g, \phi^{\text{dec}}, \phi^{\text{enc}}, \phi^{\text{prior}}$ as primary networks and refer to the components responsible for generating parameters, $\theta$ and $\omega$, as hyper networks in the following sections.

**Implementation** Following the practice of VAE, we parametrize the covariance matrices $\Sigma_t^{\mathrm{prior}}$, $\Sigma_t^{\mathrm{dec}}$ and $\Sigma_t^{\mathrm{enc}}$ as diagonal matrices. Note that $\Sigma_t^{\mathrm{prior}}$ in our model is no longer an identity matrix as in a vanilla VAE; it is the output of $\phi^{\mathrm{prior}}$ and depends on the hidden state $\mathbf{h}_{t-1}$ at the previous time step.

The recurrence model $g$ in Eq. 1 is implemented as an RNN cell, which takes as input $\mathbf{x}_t$ and $\mathbf{z}_t$ at each time step $t$ and updates the hidden states $\mathbf{h}_{t-1}$. The parameters of $g$ are generated by the hyper network $\theta(\mathbf{z}_t, \mathbf{h}_{t-1})$, as illustrated in Figure 1b. $\theta$ is also implemented using an RNN to capture the history of data dynamics, with $\mathbf{z}_t$ and $\mathbf{h}_{t-1}$ as input at each time step $t$. However, it is computationally costly to generate all the parameters of $g$ using $\theta(\mathbf{z}_t, \mathbf{h}_{t-1})$. Following the practice of previous works (Ha et al., 2016; Krueger et al., 2017), the hyper network $\theta$ maps $\mathbf{z}_t$ and $\mathbf{h}_{t-1}$ to bias and scaling vectors. The scaling vectors modify the parameters of $g$ by scaling each row of the weight matrices, routing information in the input and hidden state vectors through different channels. To better illustrate this mechanism, we exemplify the recurrence model $g$ using an RNN cell with LSTM-style update rules and gates. Let $* \in \{\mathrm{i}, \mathrm{f}, \mathrm{g}, \mathrm{o}\}$ denote the one of the four LSTM-style gates in $g$. $\mathbf{W}_*$ and $\mathbf{U}_*$ denote the input and recurrent weights of each gate in LSTM cell respectively. The hyper network $\theta(\mathbf{z}_t, \mathbf{h}_{t-1})$ outputs $\mathbf{d}_{\mathrm{i}*}$ and $\mathbf{d}_{\mathrm{h}*}$ that are the scaling vectors for the input weights $\mathbf{W}_*$ and recurrent weights $\mathbf{U}_*$ of the recurrent model $g$ in Eq. 1. The overall implementation of $g$ in Eq. 1 can be described as follows:

$$\mathbf{i}_t = \sigma\left(\mathbf{d}_{\mathrm{ii}}(\mathbf{z}_t, \mathbf{h}_{t-1}) \circ (\mathbf{W}_{\mathrm{i}}\mathbf{y}_t) + \mathbf{d}_{\mathrm{hi}}(\mathbf{z}_t, \mathbf{h}_{t-1}) \circ (\mathbf{U}_{\mathrm{i}}\mathbf{h}_{t-1})\right),$$
$$\mathbf{f}_t = \sigma\left(\mathbf{d}_{\mathrm{if}}(\mathbf{z}_t, \mathbf{h}_{t-1}) \circ (\mathbf{W}_{\mathrm{f}}\mathbf{y}_t) + \mathbf{d}_{\mathrm{hf}}(\mathbf{z}_t, \mathbf{h}_{t-1}) \circ (\mathbf{U}_{\mathrm{f}}\mathbf{h}_{t-1})\right),$$
$$\mathbf{g}_t = \tanh\left(\mathbf{d}_{\mathrm{ig}}(\mathbf{z}_t, \mathbf{h}_{t-1}) \circ (\mathbf{W}_{\mathrm{g}}\mathbf{y}_t) + \mathbf{d}_{\mathrm{hg}}(\mathbf{z}_t, \mathbf{h}_{t-1}) \circ (\mathbf{U}_{\mathrm{g}}\mathbf{h}_{t-1})\right),$$
$$\mathbf{o}_t = \sigma\left(\mathbf{d}_{\mathrm{io}}(\mathbf{z}_t, \mathbf{h}_{t-1}) \circ (\mathbf{W}_{\mathrm{o}}\mathbf{y}_t) + \mathbf{d}_{\mathrm{ho}}(\mathbf{z}_t, \mathbf{h}_{t-1}) \circ (\mathbf{U}_{\mathrm{o}}\mathbf{h}_{t-1})\right),$$
$$\mathbf{c}_t = \mathbf{f}_t \circ \mathbf{c}_{t-1} + \mathbf{i}_t \circ \mathbf{g}_t,$$
$$\mathbf{h}_t = \mathbf{o}_t \circ \tanh\left(\mathbf{c}_t\right),$$

where $\circ$ denotes the Hadamard product and $\mathbf{y}_t$ is a fusion (e.g., concatenation) of observation $\mathbf{x}_t$ and latent variable $\mathbf{z}_t$. For simplicity of notation, bias terms are ignored from the above equations. This implementation of the recurrence model in VHRNN is further illustrated in the diagram in Fig. 6 in appendix.

Another hyper network $\omega(\mathbf{z}_t, \mathbf{h}_{t-1})$ generates the parameters of the generative model in Eq. 2. It is implemented as a multilayer perceptron (MLP). Similar to $\theta(\mathbf{z}_t, \mathbf{h}_{t-1})$, the outputs are the bias and scaling vectors that modify the parameters of the decoder $\phi_{\omega(\mathbf{z}_t, \mathbf{h}_{t-1})}^{\mathrm{dec}}$.

## 4 SYSTEMATIC GENERALIZATION ANALYSIS OF VHRNN

In terms of the general functional form Eq. 1, the recurrence of VRNN and VHRNN both depend on $\mathbf{z}_t$ and $\mathbf{h}_{t-1}$, so a sufficiently large VRNN could capture the same behaviour as VHRNN in theory. However, VHRNN's structure better encodes the inductive bias that the underlying dynamics could change, that they could slightly deviate from the typical behaviour in a regime, or there could be drastic switch to a new regime. With finite training data and finite parameters, this inductive bias could lead to qualitatively different learned behaviour, which we demonstrate and analyze now.

In the spirit of Bahdanau et al. (2019), we perform a systematic generalization study of VHRNN in comparison to the VRNN baseline. We train the models on one synthetic dataset with each sequence generated by fixed linear dynamics and corrupted by heteroskedastic noise process. We demonstrate that VHRNN can disentangle the two contributions of variations and learn the different base patterns of the complex dynamics while doing so with fewer parameters. Furthermore, VHRNN can generalize to a wide range of unseen dynamics, albeit the much simpler training set.

The synthetic dataset is generated by the following recurrence equation:

$$\mathbf{x}_t = \mathbf{W}\mathbf{x}_{t-1} + \sigma_t \boldsymbol{\epsilon}_t, \tag{6}$$

where $\boldsymbol{\epsilon}_t \in \mathbb{R}^2$ is a two-dimensional standard Gaussian noise and $\mathbf{x}_0$ is randomly initialized from a uniform distribution over $[-1, 1]^2$. For each sequence, $\mathbf{W} \in \mathbb{R}^{2\times2}$ is sampled from 10 predefined random matrices $\{\mathbf{W}_i\}_{i=1}^{10}$ with equal probability; $\sigma_t$ is the standard deviation of the additive noise at time $t$ and takes value from $\{0.25, 1, 4\}$. The noise level shifts twice within a sequence; i.e., there

are exactly two $t$'s such that $\sigma_t \neq \sigma_{t-1}$. We generate 800 sequences for training, 100 sequences for validation, and 100 sequences for test using the same sets of predefined matrices. The models are trained and evaluated using FIVO as the objective. The results on the test set are almost the same as those on the training set for both VRNN and VHRNN. We also find that VHRNN shows better performance than VRNN with fewer parameters, as shown in Tab. 1, column **Test**. The size of the hidden state in RNN cells is set to be the same as the latent size for both types of models.

We further study the behavior of VRNN and VHRNN under the following systematically varied settings:

- **NOISELESS** In this setting, sequences are generated using a similar recurrence rule with the same set of predefined weights without the additive noise at each step. That is, $\sigma_t = 0$ in Eq. 6 for all time step $t$. The exponential growth of data could happen when the singular values of the underlying weight matrix are greater than 1.

- **SWITCH** In this setting, three **NOISELESS** sequences are concatenated into one, which contains regime shifts as a result. This setting requires the model to identify and re-identify the underlying pattern after observing changes.

- **RAND** In this setting, the deterministic transition matrix in Eq. 6 is set to the identity matrix (i.e., $\mathbf{W} = \mathbf{I}$), leading to long sequences of pure random walks with switching magnitudes of noise. The standard deviation of the additive noise randomly switches up to 3 times within $\{0.25, 1, 4\}$ in one sequence.

- **LONG** In this setting, we generate extra-long **NOISELESS** sequences with twice the total number of steps using the same set of predefined weights. The data scale can exceed well beyond the range of training data when exponential growth happens.

- **ZERO-SHOT** In this setting, **NOISELESS** sequences are generated such that the training data and test data use different sets of weight matrices.

- **ADD** In this setting, sequences are generated by a different recurrence rule: $\mathbf{x}_t = \mathbf{x}_{t-1} + \mathbf{b}$, where $\mathbf{b}$ and $\mathbf{x}_0$ are uniformly sampled from $[0, 1]^2$.

Tab. 1 illustrates the experimental results. We can see that VRNN model, depending on model complexity, either underfits the original data generation pattern (**Test**) or fails to generalize to more complicated settings. In contrast, the VHRNN model does not suffer from such problems and uniformly outperforms VRNN models under all settings. To qualitatively study the behavior of VHRNN and VRNN, we consider a VRNN with a latent dimension of 8 and a VHRNN with a latent dimension of 4 and make the following observations:

Table 1: Evaluation results on synthetic datasets.

| Model | Z dim. | Param. | FIVO estimated log likelihood per time step | | | | | | |
|-------|--------|--------|------|-----------|--------|------|------|-----------|------|
| | | | **Test** | **NOISELESS** | **SWITCH** | **RAND** | **LONG** | **ZERO-SHOT** | **ADD** |
| VRNN | 8 | 2612 | $-5.43$ | $-2.50$ | $-334173$ | $-5.02$ | $-1033348$ | $-3.64$ | $-3.57$ |
| VRNN | 6 | 1516 | $-5.80$ | $-3.66$ | $-19735$ | $-5.24$ | $-27200$ | $-4.39$ | $-5.09$ |
| VRNN | 4 | 716 | $-7.88$ | $-5.25$ | $-7.85$ | $-6.81$ | $-5777$ | $-5.35$ | $-8.67$ |
| VHRNN | 4 | 1568 | $-4.68$ | $-2.08$ | $-4.27$ | $-3.91$ | $-3005$ | $-2.57$ | $-2.62$ |

**Dynamic Regime Identification and Re-identification** Fig. 2 shows a sample sequence under the **NOISELESS** setting. VRNN has high KL divergence between the prior and the variational posterior most of the time. In contrast, VHRNN has a decreasing trend of KL divergence while still making accurate mean reconstruction as it observes more data. As the KL divergence measures the discrepancy between prior defined in Eq. 3 and the posterior that has information from the current observation, simultaneous low reconstruction and low KL divergence means that the prior distribution would be able to predict with low errors as well, indicating that the correct underlying dynamics model has likely been utilized. This trend even generalizes to settings with sources of variation unseen in the training data, namely **ZEROSHOT** and **ADD**. We speculate that this trend implies the model's ability to identify the underlying data generation pattern in the sequence. The decreasing trend is especially apparent when a sudden and big change in scale happens. We hypothesize that larger changes in scale can better help our model, VHRNN, identify the underlying data generation process because our model is trained on sequential data generated with compound noise. The observation further corroborates our conjecture that the KL divergence would rise again once the sequence

switches from one underlying weight to another, as shown in Fig. 3. It is worth noting that the KL increase happens with some latency after the sequence switches in the **SWITCH** setting as the model reacts to the change and tries to reconcile with the prior belief of the underlying regime in effect.

**Uncertainty Identification** Fig. 4 shows that the predicted log-variance of VHRNN can more accurately reflect the change of noise levels under the **RAND** setting than VRNN. VHRNN can also better handle uncertainty than VRNN in the following two situations. As shown in Fig. 3f, VHRNN can more aggressively adapt its variance prediction based on the scale of the data than VRNN. It keeps its predicted variance at a low level when the data scale is small and increases the value when the scale of data becomes large. VHRNN makes inaccurate mean prediction relatively far from the target value when the switch of underlying generation dynamics happens in the **SWITCH** setting. The switch of the weight matrix is another important source of uncertainty. We observe that VHRNN would also make a large log-variance prediction in this situation, even the scale of the observation is small. Aggressively increasing its uncertainty about the prediction when a switch happens avoids VHRNN model from paying high reconstruction cost as shown by the second spike in Fig. 3f. This increase of variance prediction also happens when exponential becomes apparent in setting **LONG** and the scale of observed data became out of the range of the training data. Given the large scale change of the data, such flexibility to predict large variance is key for VHRNN to avoid paying large reconstruction cost.

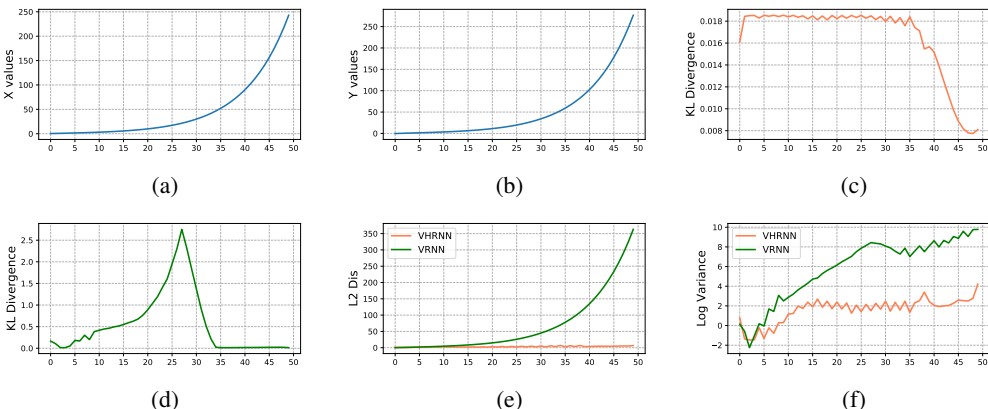

Figure 2: Qualitative study of VRNN and VHRNN under the **NOISELESS** setting. (a) and (b) show the values of concatenated data at each time step. (c) shows the KL divergence between the variational posterior and the prior of the latent variable at each time step for VHRNN. (d) shows the KL divergence for VRNN. (e) shows L2 distance between the predicted mean values by VHRNN and VRNN and the target. (f) shows the predicted log-variance of the output distribution for VRNN and VHRNN.

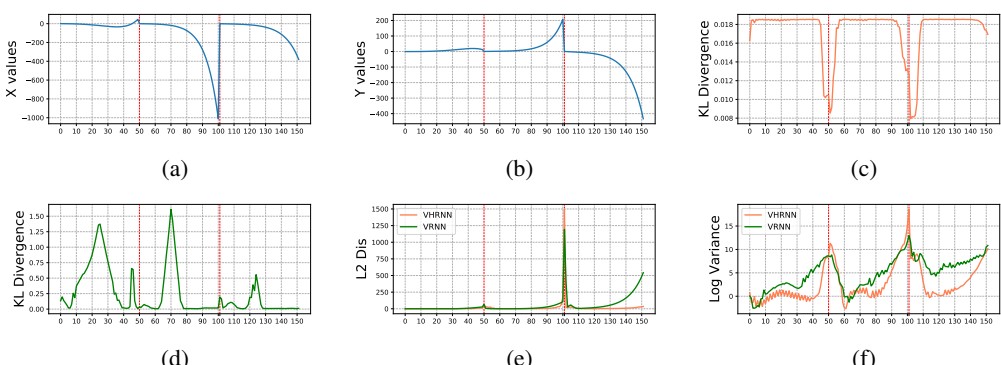

Figure 3: Qualitative study of VRNN and VHRNN under the **SWITCH** setting. The layout of subfigures is the same as Fig. 2. Vertical red lines indicate time steps when regime shift happen.

These two advantages of VHRNN over VRNN not only explain the better performance of VHRNN on the synthetic data but also are critical to RNNs' ability to model real-world data with large variations

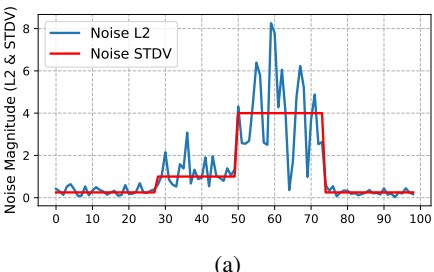 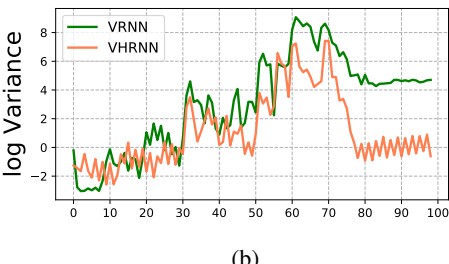

(a)               (b)

Figure 4: Qualitative study of VRNN and VHRNN under the **RAND** setting. (a) shows the L2 norm and standard deviation of the additive noise at each time step. (b) shows the log-variance of the output distribution for VRNN and VHRNN.

both across and within sequences. Examples under other settings showing the above properties are deferred to the Appendix.

## 5 EXPERIMENTS ON REAL-WORLD DATA

We experiment with the VHRNN model on several real-world datasets and compare it against VRNN model. VRNN trained and evaluated using FIVO (Maddison et al., 2017) demonstrates the state-of-the-art performance on various sequence modeling tasks. Our experiments demonstrate the superior parameter-performance efficiency and generalization ability of VHRNN over VRNN. All the models are trained using FIVO (Maddison et al., 2017) and we report FIVO per step when evaluating models. Two polyphonic music dataset are considered: JSB Chorale and Piano-midi.de (Boulanger-Lewandowski et al., 2012). We also train and test our models on a financial time series data and the HT Sensor dataset (Huerta et al., 2016), which contains sequences of sensor readings when different types of stimuli are applied in an environment during experiments. We also trained and evaluated the HyperLSTM model without latent variables proposed by Ha et al. (2016) on a few of the datasets above. The results and comparisons are deferred to the appendix.

For the VRNN model, we use a single-layer LSTM and set the dimension of the hidden state to be the same as the latent dimension. For the VHRNN model, $\theta$ in Eq. 1 is implemented using a single-layer LSTM to generate weights for the recurrence module in the primary networks. We use an RNN cell with LSTM-style gates and update rules for the recurrence module $g$ in our experiments. The hidden state sizes of both the primary network and hyper network are the same as the latent dimension. A linear transformation directly maps the hyper hidden state to the scaling and bias vectors in the primary network. More details on the architectures of encoder, generation and prior networks are elaborated in the appendix.

**Polyphonic Music** The JSB Chorale and Piano-midi.de are music datasets (Boulanger-Lewandowski et al., 2012) with complex patterns and large variance both within and across sequences. The datasets are split into the standard train, validation, and test sets. More details on data preprocessing, training and evaluation setup are deferred to the appendix.

We report the FIVO per time step of VHRNNs and VRNNs and their parameter counts in Fig. 5a and Fig. 5b. The results show that VHRNNs have better performance and parameter efficiency. The number of parameters and FIVO per time step of each model are plotted in the figures, and the latent dimension is also annotated. The parameter-performance plots show that the VHRNN model has uniformly better performance than VRNN with a comparable number of parameters. The best FIVO achieved by VHRNN on JSB dataset is $-6.76$ (VHRNN-14) compared to $-6.92$ for VRNN (VRNN-32), which requires close to one third more parameters. This best VRNN model is even worse than the smallest VHRNN model we have evaluated. It is also observed that VHRNN is less prone to overfitting and has better generalization ability than VRNN when the number of parameters keeps growing. Similar trends can be seen on the Piano-midi.de dataset in Fig. 5b. We also find that the better performance of VHRNN over VRNN can generalize to the scenario where we replace LSTM with Gated Recurrent Unit (GRU). Experimental results using GRU implementation are deferred to the appendix.

**Stock** Financial time series data, such as daily prices of stocks, are highly volatile with large noise. The market volatility is affected by many external factors and can experience tremendous changes

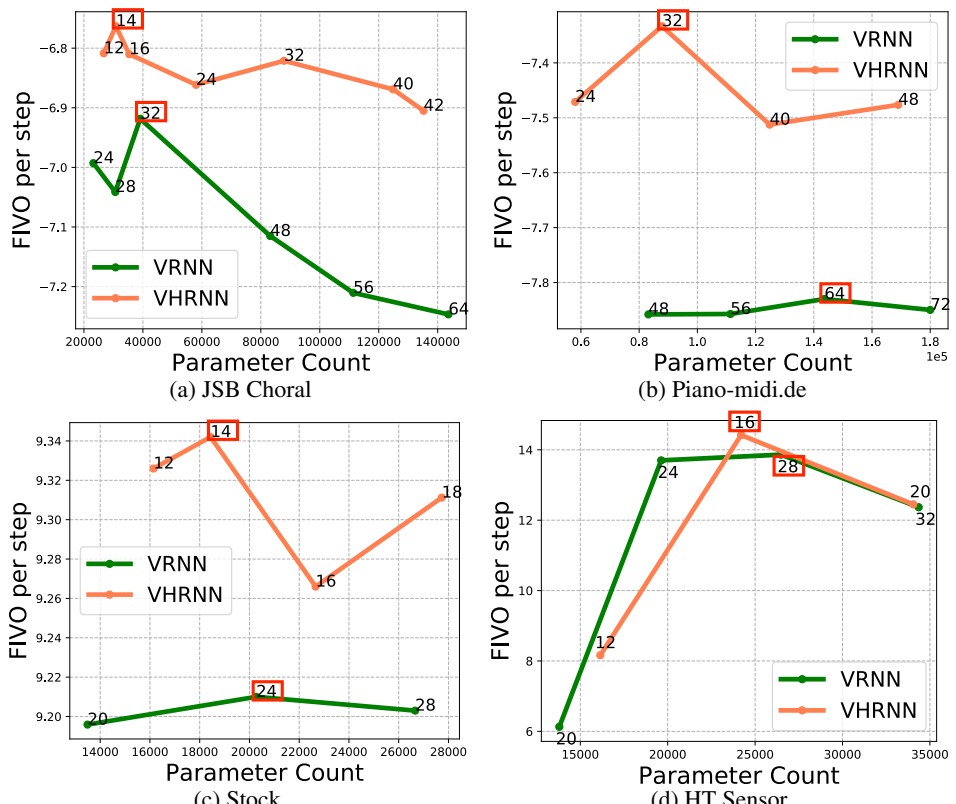

Figure 5: VRNN and VHRNN parameter-performance comparison.

in a sudden. To test the models' ability to adapt to different volatility levels and noise patterns, we compare VHRNN and VRNN on stock price data collected in a period when the market went through rapid changes. The data are collected from 445 stocks in the S&P500 index in 2008 when a global financial crisis happened. The dataset contains the opening, closing, highest and lowest prices, and volume on each day. The networks are trained on sequences from the first half of the year and tested on sequences from the second half, during which the market suddenly became significantly more volatile due to the financial crisis.

The evaluation results are shown in Fig. 5c. The plot shows that VHRNN models consistently outperform VRNN models regardless of the latent dimension and number of parameters. The results indicate that VHRNN can have better generalizability to sequential data in which the underlying data generation pattern suddenly shifts even if the new dynamics are not seen in the training data.

**HT Sensor** The comparison is also performed on a dataset with less variation and simpler patterns than the previous datasets. The HT Sensor dataset contains sequences of gas, humidity, and temperature sensor readings in experiments where some stimulus is applied after a period of background activity (Huerta et al., 2016). There are only two types of stimuli in the experiments: banana and wine. In some sequences, there is no stimulus applied, and they only contain readings under background noise. Experimental results on HT Sensor dataset are shown in Fig. 5d.

It is observed that VHRNN has comparable performance as VRNN on the HT Senor Dataset when using a similar number of parameters. For example, VHRNN achieves a FIVO per time step of 14.41 with 16 latent dimensions and 24200 parameters, while VRNN shows slightly worse performance with 28 latent dimensions and approximately 26000 parameters. When the number of parameters goes slightly beyond 34000, the FIVO of VHRNN decays to 12.45 compared to 12.37 of VRNN.

# 6 ABLATION STUDY

We further investigate the effects of hidden state and latent variable on the performance of variational hyper RNN in the following two aspects: the dimension of the latent variable and the contributions by hidden state and latent variable as inputs to hyper networks.

**Latent Dimension** In previous experiments on real-world datasets, the latent dimension and hidden state dimension are set to be the same for each model. This causes VHRNN to have significantly more parameters than a VRNN when using the same latent dimension. To eliminate the effects of the difference in model size, we allow the latent dimension and hidden state dimension to be different. We also reduce the hidden layer size of the hyper network that generates the weight of the decoder. These changes allow us to compare VRNN and VHRNN models with the same latent dimension and a similar number of parameters. The results on JSB Chorale datasets are presented in Tab. 2 in which we denote latent dimension by Z dim. We observe that VHRNNs always have better FIVO with the same latent dimensions than VRNNs. The results show that the superior performance of VHRNN over VRNN does not stem from smaller latent dimension when using the comparable number of parameters.

**Inputs to the Hyper Networks** We retrain and evaluate the performance of VHRNN models on JSB Chorale dataset and the synthetic sequences when feeding the latent variable only, the hidden state only, or both to the hyper networks. The results are shown in Tab. 3. It is observed that VHRNN has the best performance and generalization ability when it takes the latent variable as its only input. Relying on the primary network's hidden state only or the combination of latent variable and hidden state leads to worse performance. When the dimension of the hidden state is 32, VHRNN only taking the hidden state as hyper input suffers from over-parameterization and has worse performance than VRNN with the same dimension of the hidden state. On the test set of synthetic data, VHRNN obtains the best performance when it takes both hidden state and latent variable as inputs. We surmise that this difference is due to the fact that historical information is critical to determine the underlying recurrent weights and current noise level for synthetic data. However, the ablation study on both datasets shows the importance of the sampled latent variable as an input to the hyper networks. Therefore, both hidden state and latent variable are used as inputs to hyper networks on other datasets for consistency.

## 7 CONCLUSION

In this paper, we introduce the variational hyper RNN (VHRNN) model, which can generate parameters based on the observations and latent variables dynamically. Such flexibility enables VHRNN to better model sequential data with complex patterns and large variations within and across samples than VRNN models that use fixed weights. VHRNN can be trained with the existing off-the-shelf variational objectives. Experiments on synthetic datasets with different generating patterns show that VHRNN can better disentangle and identify the underlying dynamics and uncertainty in data than VRNN. We also demonstrate the superb parameter-performance efficiency and generalization ability of VHRNN on real-world datasets with different levels of variability and complexity.

Table 2: VRNN and VHRNN with same latent dimensions.

| Model | Z dim. | Hidden dim. | Hyper size | Param. | FIVO |
|-------|--------|-------------|------------|--------|------|
| VRNN | 24 | 24 | - | 23k | $-7.04$ |
| | 28 | 28 | - | 31k | $-6.99$ |
| | 32 | 32 | - | 39k | $-6.91$ |
| VHRNN | 24 | 12 | 16 | 24k | $-6.92$ |
| | 28 | 14 | 18 | 31k | $-6.73$ |
| | 32 | 16 | 20 | 39k | $-6.70$ |

Table 3: Ablation study with different hyper network inputs.

| Dataset | Z dim. | Hyper Input | FIVO |
|---------|--------|-------------|------|
| JSB | **14** | **latent only** | $-$**6.68** |
| | 14 | hidden only | $-6.71$ |
| | 14 | latent+hidden | $-6.76$ |
| | **32** | **latent only** | $-$**6.76** |
| | 32 | hidden only | $-7.03$ |
| | 32 | latent+hidden | $-6.82$ |
| Synthetic Test | 4 | latent only | $-5.01$ |
| | 4 | hidden only | $-4.79$ |
| | **4** | **latent+hidden** | $-$**4.68** |

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

## A    VHRNN Recurrence Model Implementation using LSTM Cell

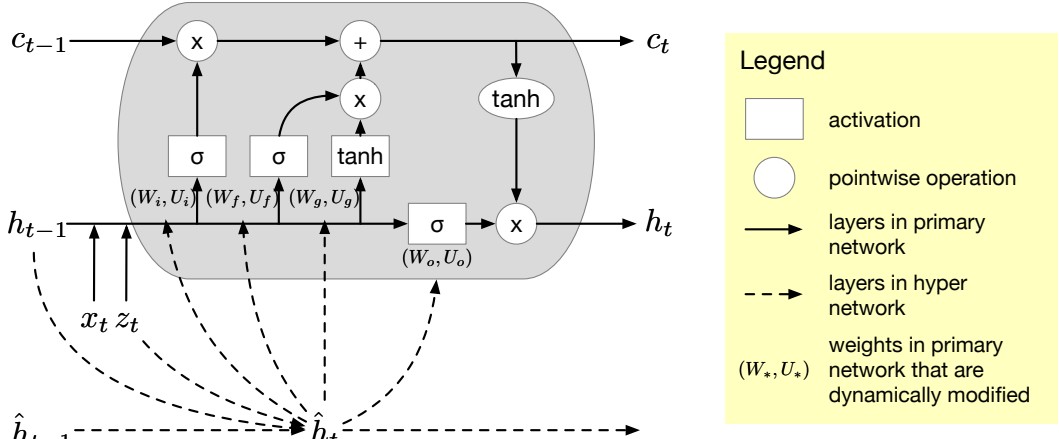

Figure 6: Implementation of the recurrence model in VHRNN using LSTM cell.

## B    Qualitative Study of VRNN and VHRNN under **ADD**, **ZERO-SHOT**, **LONG** Settings on Synthetic Datasets

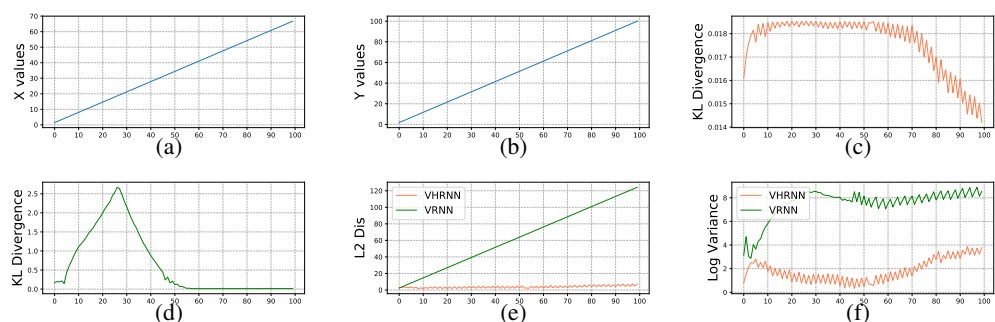

Figure 7: Qualitative study of VRNN and VHRNN under the **ADD** setting. (a) and (b) show the values of concatenated data at each time step. (c) shows the KL divergence between the variational posterior and the prior of the latent variable at each time step for VHRNN. (d) shows the KL divergence for VRNN. (e) shows L2 distance between the predicted mean values by VHRNN and VRNN and the target. (f) shows the predicted log-variance of the output distribution for VRNN and VHRNN.

Fig. 7 and Fig. 8 show the qualitative study results of VHRNN and VRNN under the **ADD** and **ZERO-SHOT** settings. We can see that the KL divergence of VHRNN model decreases as it observes more data. Meanwhile the mean predictions by VHRNN stay relatively close to the actual target value as shown in Fig. 7e and Fig. 8e. The prediction are especially accurate in the **ADD** setting as Fig. 7e shows. The results demonstrate VHRNN's ability to identify system dynamics can generalize to unseen data generation patterns. By contrast, we does not see any trend of variational RNN that indicates it is capable of doing dynamic regime identification.

Fig. 9 illustrates the qualitative study of VHRNN and VRNN under the **LONG** setting. The magnitude of the data grows rapidly in such setting due to exponential growth and it is well beyond the scale of training data. We can see both VRNN and VHRNN make very **inaccurrate** mean predictions that are far from the target values as Fig. 9e shows. However, VHRNN pays smaller reconstruction cost than VRNN by also making large predictions of variance. This setting demonstrates a special case in which VHRNN has better ability to handle uncertainty in data than vanilla variational RNN.

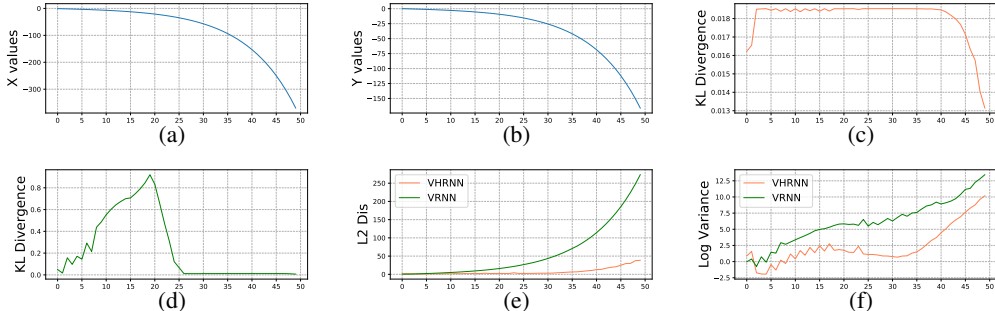

Figure 8: Qualitative study of VRNN and VHRNN under the **ZERO-SHOT** setting. The layout of subfigures is the same as Fig. 7.

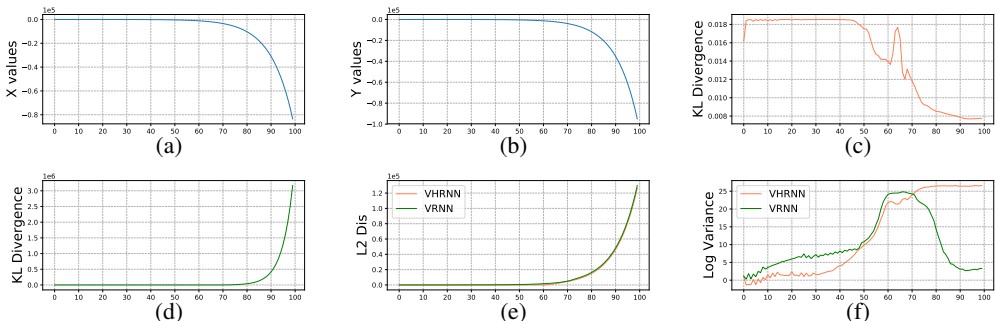

Figure 9: Qualitative study of VRNN and VHRNN under the **LONG** setting. The layout of subfigures is the same as Fig. 7. Fig. 9a, 9b, 9d, 9e use scientific notations for the value of Y axis.

## C   VRNN AND VHRNN IMPLEMENTATION DETAILS

**Encoder Network and Prior Network** The architecture of the encoder in Equation 5 is the same in VHRNN and VRNN. For synthetic datasets, the encoder is implemented by a fully-connected network with two hidden layers; each hidden layer has the same number of units as the latent variable dimension. For other real-world datasets, we use a fully-connected network with one hidden layer. The number of units is also the same as the latent dimension. The prior network is implemented by a similar architecture as the encoder, with the only difference being the dimension of inputs.

**Decoder Network** For VHRNN models, we use fully-connected hyper networks with two hidden layers for synthetic data and fully-connected hyper networks with one hidden layer for other datasets as the decoder networks. The number of units in each hidden layer is also the same as the latent variable defined in Equation 2. For each layer of the hyper networks, the weight scaling vector and bias is generated by an two-layer MLP. The hidden layer size of this MLP is 8 for synthetic dataset and 64 for real-world datasets. For VRNN models, we use plain feed-forward networks for decoder. The number of hidden layers and units in the hidden layer are determined in the same way as VHRNN models.

**Observation and Latent Variable Encoding Network** For fair comparison with VRNN Chung et al. (2015), the latent variable and observations are encoded by a network (different from the encoder in Equation 5) before being fed to the recurrence network and encoder. The latent and observation encoding networks have the same architecture except for the input dimension in each experiment setting. For synthetic datasets, the encoding network is implemented by a fully-connected network with two hidden layers. For real-world datasets, we use a fully-connected network with one hidden layer. The number of units in each hidden layer is the same as the dimension of latent variable in that experiment setting.

## D   REAL-WORLD DATASETS PREPROCESSING DETAILS

**Polyphonic Music** Each sample in the polyphonic music datasets, JSB Chorale and Piano-midi.de is represented as a sequence of 88-dimensional binary vectors. The data are preprocessed by mean-centering along each dimension per dataset.

**Stock** We randomly select 345 companies and use their daily stock price and volume in the first half of 2008 to obtain training data. We another 50 companies' data in the second half of 2008 to generate validation set and get the test set from the remaining 50 companies during the second half of 2008. The sequences are first preprocessed by taking log ratio of the values between consecutive days. Each sequence has a fixed length of 125. The log ratio sequences are normalized using the mean and standard deviation of the training set along each dimension.

**HT Sensor** The HT Sensor dataset collects readings from 11 sensors under certain stimulus in an experiment. The readings of the sensors are recorded at a rate of once per second. We segment a sequence of 3000 seconds every 1000 seconds in the dataset and downsample the sequence by a rate of 30. Each sequence we obtained has a fixed length of 100. The types of sequences include pure background noise, stimulus before and after background noise and stimulus between two periods of background noise. The data are normalized to zero mean and unit variance along each dimension. We use 532 sequences for training, 68 sequences for validation and 74 sequences for testing.

## E   TRAINING AND EVALUATION DETAILS ON REAL-WORLD DATASETS.

For all the real-world data, the models, both VRNN and VHRNN, are trained with batch size of 4 and particle size of 4. When evaluating the models, we use particle size of 128 for polyphonic music datasets and 1024 for Stock and HT Sensor datasets.

## F   VHRNN AND VRNN HIDDEN UNIT VERSUS PERFORMANCE COMPARISON

We also compare VHRNN and VRNN by plotting the models' performance against their number of hidden units. The models we considered here are the same as the models presented in Fig. 5: We use a single-layer LSTM model for the RNN part; the dimension of LSTM's hidden state is the same as the latent dimension. It's worth noting that as VHRNN uses two LSTM models, one primary network and one hyper network. Therefore, the number of hidden units in an VHRNN model is twice the number of latent dimension. We can see that VHRNN also dominates the performance of VRNN with a similar or fewer number of hidden units in most of the settings. Furthermore, the fact that VHRNN almost always outperforms VRNN for all parameter or hidden unit sizes precisely shows the superiority of the new architecture. The results from Fig. 5 and Fig. 10 are consolidated in into Tab. 4.

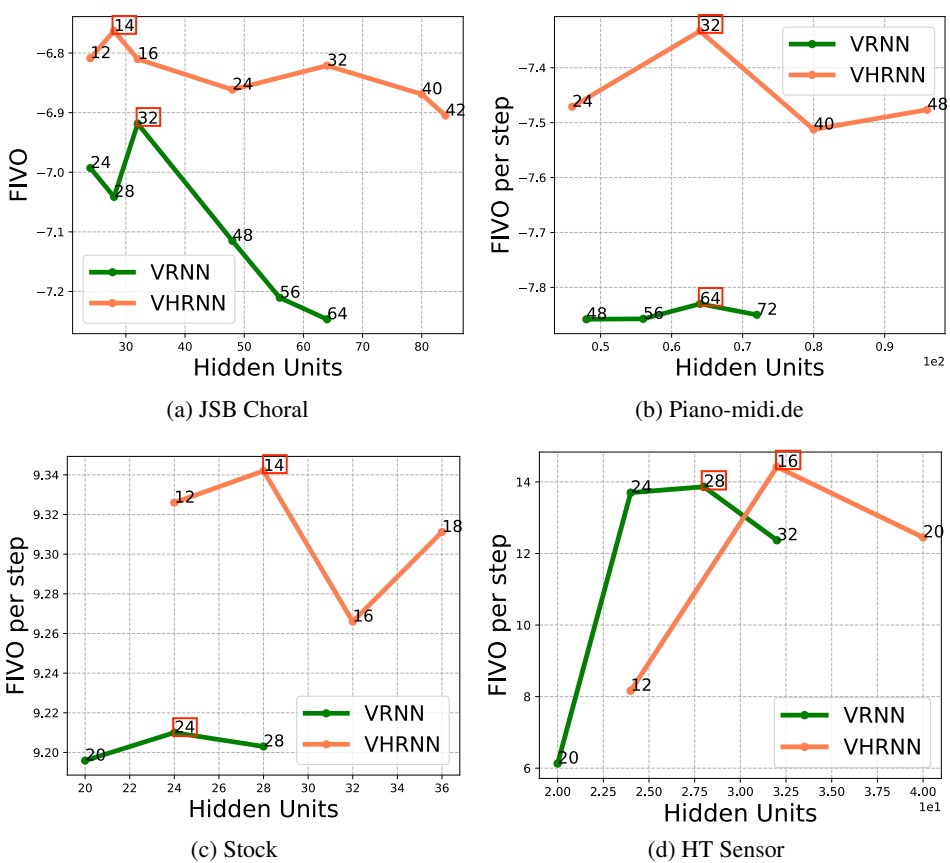

(a) JSB Choral

(b) Piano-midi.de

(c) Stock

(d) HT Sensor

Figure 10: VRNN and VHRNN Hidden Units vs Performance comparison.

Table 4: VRNN (left) and VHRNN (right) on Real-World Datasets.

| Dataset | Z dim. | Hidden Unit | Param. | FIVO | Dataset | Z dim. | Hidden Unit | Param. | FIVO |
|---|---|---|---|---|---|---|---|---|---|
| JSB | 64 | 64 | 144k | −7.25 | JSB | 42 | 84 | 135k | −6.90 |
| | 56 | 56 | 111k | −7.21 | | 40 | 80 | 125k | −6.87 |
| | 48 | 48 | 83k | −7.12 | | 32 | 64 | 88k | −6.82 |
| | **32** | **32** | **39k** | **−6.92** | | 24 | 48 | 58k | −6.86 |
| | 28 | 28 | 31k | −6.99 | | 16 | 32 | 35k | −6.81 |
| | 24 | 24 | 23k | −7.04 | | **14** | **28** | **31k** | **−6.76** |
| | | | | | | 12 | 24 | 27k | −6.80 |
| Piano | 80 | 80 | 220k | −7.85 | Piano | 48 | 96 | 169k | −7.48 |
| | 72 | 72 | 180k | −7.85 | | 40 | 80 | 125k | −7.51 |
| | **64** | **64** | **144k** | **−7.83** | | **32** | **64** | **88k** | **−7.33** |
| | 56 | 56 | 111k | −7.86 | | 24 | 48 | 58k | −7.47 |
| | 48 | 48 | 83k | −7.86 | | | | | |
| Stock | 20 | 20 | 13k | 9.20 | Stock | 12 | 24 | 16k | 9.33 |
| | **24** | **24** | **20k** | **9.21** | | **14** | **28** | **18k** | **9.34** |
| | 28 | 28 | 27k | 9.20 | | 16 | 32 | 23k | 9.27 |
| | | | | | | 18 | 36 | 28k | 9.31 |
| HT Sensor | 20 | 20 | 14k | 6.13 | HT Sensor | 12 | 24 | 16k | 8.16 |
| | **24** | **24** | **20k** | **13.70** | | **16** | **32** | **24k** | **14.41** |
| | 28 | 28 | 26k | 12.37 | | 20 | 40 | 34k | 12.45 |
| | 32 | 32 | 34k | 12.69 | | | | | |

# G  VHRNN AND VRNN PERFORMANCE-PARAMETER COMPARISON USING GRU ON JSB CHORALE DATASET

Fig. 11 shows the parameter performance plots of VHRNN and VRNN using GRU implementation on the JSB Chorale dataset. VHRNN models consistently outperform VRNN models under all settings.

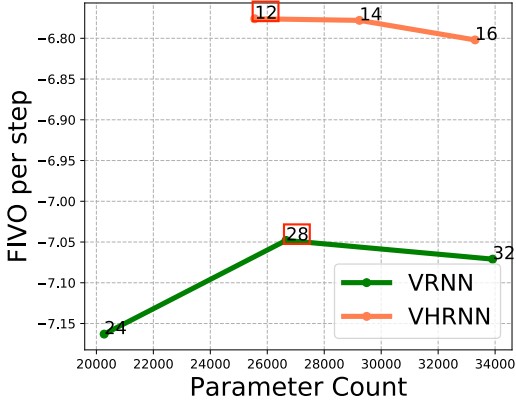

Figure 11: VRNN and VHRNN arameter-performance comparison using GRU implementation on JSB Chorale dataset.

# H  VHRNN AND HYPERLSTM PERFORMANCE COMPARISON

We compare our VHRNN models using LSTM cell with the HyperLSTM models proposed in HyperNetworks Ha et al. (2016) on JSB Chorale and Stock datasets. Compared with VHRNN, HyperLSTM does not have latent variables. Therefore, it does not have an encoder or decoder either. Our implementation of HyperLSTM resembles the recurrence model of VHRNN defined in Equation 6. At each time step, HyperLSTM model predicts the output distribution by mapping the RNN's hidden state to the parameters of binary distributions for JSB Chorale dataset and a mixture of Gaussian for Stock dataset. We consider 3 and 5 as the number of components in the Gaussian

mixture distribution. HyperLSTM models are trained with the same batch size and learning rate as VHRNN models.

We show the parameter-performance comparison between VHRNN, VRNN and HyperLSTM models in Fig. 12. The number of components used by HyperLSTM for Stock dataset is 5 in the plot. Since HyperLSTM models do not have latent variable, the indicator on top of each point shows the number of hidden units in each model for all the three of them. The number of hidden units for HyperLSTM model is also twice the dimension of hidden states as HyperLSTM has two RNNs, one primary and one hyper. We report FIVO for VHRNN and VRNN models and exact log likelihood for HyperLSTM models. Even though FIVO is a lower-bound of log likelihood, we can see that the performance of VHRNN completely dominates HyperLSTM no matter what number of hidden units we use. Actually, the performance of HyperLSTM is even worse than VRNN models which do not even have hyper networks. The results indicates the importance of modeling complex time-series data.

We also show the hidden-units-performance comparison between VHRNN and VRNN in Fig. 13. The comparison shows similar results.

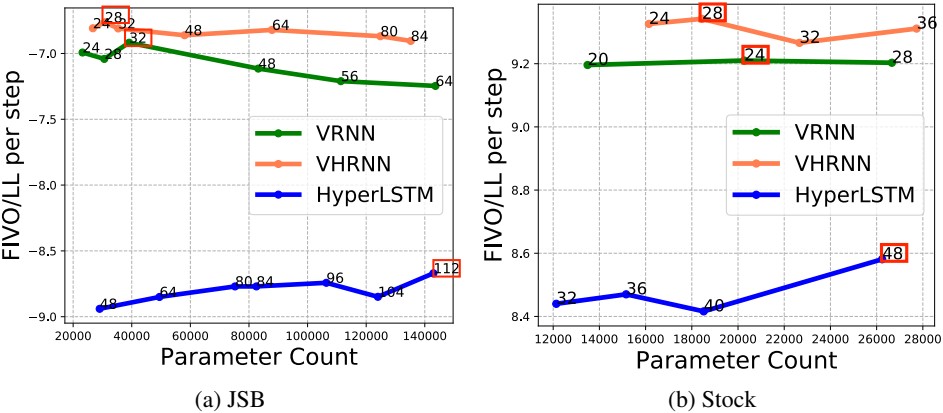

Figure 12: Parameter vs Performance comparison among VHRNN, VRNN and HyperLSTM. We report FIVO for VHRNN and VRNN, and exact log likelihood for HyperLSTM. We use a Gaussian mixture distribution of 5 components in the HyperLSTM model to estimate log likelihood in Stock dataset. The indicator above each point shows the number of hidden units in the model.

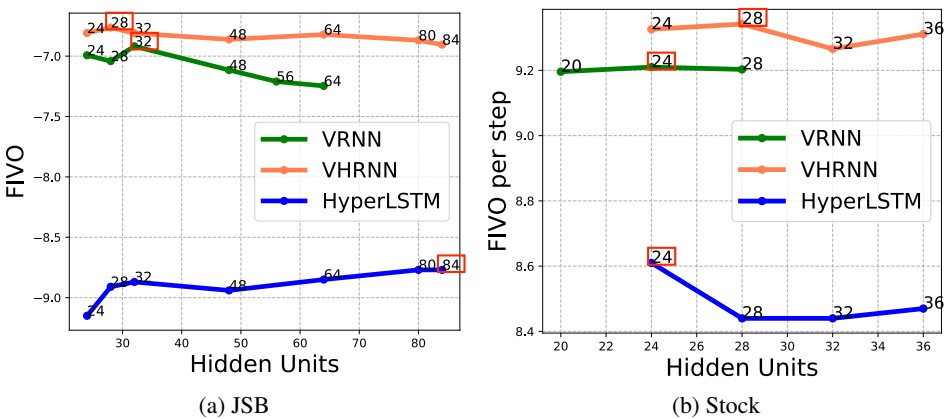

Figure 13: Hidden units vs Performance comparison among VHRNN, VRNN and HyperLSTM. We report FIVO for VHRNN and VRNN, and exact log likelihood for HyperLSTM. We use a Gaussian mixture distribution of 5 components in the HyperLSTM model to estimate log likelihood in Stock dataset. The indicator above each point shows the number of hidden units in the model.

Complete experiment results of HyperLSTM models on the two datasets are shown in Tab. 5.

Table 5: HyperLSTM Performance in Log Likelihood

| Dataset | Hidden Units | Gauss. Components | Param. | Log Likelihood |
|---|---|---|---|---|
| JSB | 24 | N.A. | 9k | $-9.15$ |
| | 28 | N.A. | 11k | $-8.91$ |
| | 32 | N.A. | 14k | $-8.87$ |
| | 48 | N.A. | 30k | $-8.94$ |
| | 64 | N.A. | 49k | $-8.85$ |
| | 80 | N.A. | 75k | $-8.77$ |
| | 84 | N.A. | 83k | $-8.77$ |
| | 96 | N.A. | 106k | $-8.74$ |
| | 104 | N.A. | 124k | $-8.85$ |
| | **112** | N.A. | **143k** | $\mathbf{-8.67}$ |
| Stock | 24 | 3 | 7k | 8.28 |
| | 28 | 3 | 9k | 8.33 |
| | **32** | **3** | **12k** | **8.42** |
| | 36 | 3 | 15k | 8.35 |
| | 40 | 3 | 18k | 8.29 |
| | 48 | 3 | 26k | 7.91 |
| | **24** | **5** | **7k** | **8.61** |
| | 28 | 5 | 9k | 8.44 |
| | 32 | 5 | 12k | 8.44 |
| | 36 | 5 | 15k | 8.47 |
| | 40 | 5 | 19k | 8.42 |
| | 48 | 5 | 26k | 8.58 |

## I  VHRNN WITHOUT TEMPORAL STRUCTURE IN HYPER NETWORKS

Using an RNN to generate the parameters of another RNN has been studied in HyperNetworks Ha et al. (2016) and delivers promising performance. It also seems like a natural choice as the hidden state of the primary RNN can represent the history of observed data while the hidden state of the hyper RNN can track the history of data generation dynamics. However, it is still intriguing to study other design choices that do not have the recurrence structure in the hyper networks for VHRNN. As an ablation study, we experimented with VHRNN models that replace the RNN with a three-layer feed-forward network as the hyper network $\theta$ for the recurrence model $g$ as defined in Equation 6. We keep the other components of VHRNN unchanged on JSB Chorale, Stock and the synthetic dataset. The evaluation results using FIVO are presented in Tab. 6 and systematic generalization study results on the synthetic dataset are shown in Tab. 7. We denote the original VHRNN with recurrence structure in $\theta$ as VHRNN-RNN and the variant without the recurrence structure as VHRNN-MLP.

As we can see, given the same latent dimension, VHRNN-MLP models have more parameters than VHRNN-RNN models. VHRNN-MLP can have slightly better performance than VHRNN-RNN in some cases but it performs worse than VHRNN-RNN in more settings. The performance of VHRNN-MLP also degrades faster than VHRNN-RNN on the JSB Chorale dataset as we increase the latent dimension. Moreover, systematic generalization study on the synthetic dataset also shows that VHRNN-MLP has worse performance than VHRNN-RNN no matter in the test setting or in the systematically varied settings.

Table 6: Parameter-Performance Comparison of VHRNN-RNN and VHRNN-MLP

| Dataset | Z dim. | VHRNN-MLP Param. | VHRNN-RNN Param. | VHRNN-MLP FIVO | VHRNN-RNN FIVO |
|---|---|---|---|---|---|
| JSB | 42 | 146k | 135k | −7.02 | **−6.90** |
| | 40 | 135k | 125k | −7.08 | **−6.87** |
| | 32 | 94k | 88k | −7.13 | **−6.82** |
| | 24 | 62k | 58k | **−6.83** | −6.86 |
| | 16 | 37k | 35k | **−6.76** | −6.81 |
| | 14 | 32k | 31k | **−6.73** | −6.76 |
| | 12 | 28k | 27k | −6.83 | **−6.80** |
| Stock | 12 | 16k | 16k | **9.47** | 9.33 |
| | 14 | 19k | 18k | 9.21 | **9.34** |
| | 16 | 24k | 23k | **9.31** | 9.27 |
| | 18 | 29k | 28k | 9.19 | **9.31** |
| Syn-Test | 4 | 1676 | 1568 | −5.13 | **−4.68** |

Table 7: Systematic Generalization Study on VHRNN-RNN and VHRNN-MLP

| Model | Z dim. | Param. | FIVO estimated log likelihood per time step | | | | | | |
|---|---|---|---|---|---|---|---|---|---|
| | | | Test | NOISELESS | SWITCH | RAND | LONG | ZERO-SHOT | ADD |
| VHRNN-MLP | 4 | 1676 | −5.13 | −2.73 | −5.22 | −4.12 | −604937 | −2.94 | −3.84 |
| VHRNN-RNN | 4 | 1568 | **−4.68** | **−2.08** | **−4.27** | **−3.91** | **−3005** | **−2.57** | **−2.62** |

