# OpenReview forum: "Variational Hyper RNN for Sequence Modeling"
_ICLR.cc/2020/Conference — Reject_

### Official Review · AnonReviewer3 · 2019-10-23
**Official Blind Review #3**

**Rating:** 6

**Review:**

This paper proposes the variational hyper RNN (VHRNN), which extends the previous variational RNN (VRNN) by learning the parameters of RNN using a hyper RNN. VRHNN is tested and compared with VRNN on synthetic and real datasets. The authors report superior performance parameter efficiency over VRNN.

The performance of VHRNN is promising and certainly better than the previous VRNN for some applications. However, the VHRNN is constructed by a straight-forward combination of existing techniques and hence the technical contribution of this paper is marginal.

Although Section 4 is entitled as systematic generalization analysis of VHRNN, the reported results are only for the specific structures of VHRNN and VRNN. Isn’t it useless to present results for the VRNN with a latent dimension of 4, at least as a sanity check?

Fig. 2 and the texts referring to it discuss the KL divergence between the prior and the variational posterior. While the FIBO is mainly used as the objective in this paper, is the ELBO enough if the authors care the simultaneous low reconstruction error and low KL divergence?

It is unclear and explained little if the comparison using parameter count is fair for VHRNN and VRNN since they have different structures.

It would be nicer to discuss for which kind of time-series VRNN is enough.

Minor comments:
The caption of Figure 1 is too close to the main texts.
Eq. (4) is overlapping with texts.
Can the equations at the bottom of p.3 be explained with an illustration?


**Experience Assessment:**

I have read many papers in this area.

**Review Assessment: Checking Correctness Of Derivations And Theory:**

I assessed the sensibility of the derivations and theory.

**Review Assessment: Checking Correctness Of Experiments:**

I assessed the sensibility of the experiments.

**Review Assessment: Thoroughness In Paper Reading:**

I made a quick assessment of this paper.

---

> ### Author Response · Authors · 2019-11-10
> **Response to AnonReviewer3**
>
> We would like to thank the reviewer for providing thoughtful comments and suggestions on improving the presentation of the paper. The suggested improvements of the paper’s presentation will be addressed in the next updated version of the paper.
>
> The reviewer acknowledged that VHRNN models show better performance than the VRNN models by combining the VHRNN with hyper networks. However, the motivation of our work is to better handle complex variabilities within and across sequences in time-series data by dynamically changing the model’s weights based on observations. Hypernetwork is a commonly adopted class of networks with dynamically generated weights. We connect dynamic weight generation with hypernetwork with variational Bayes method for sequence modeling to fulfill this task. We trained our model with the FIVO objective because it is based on the well-studied particle filtering algorithm with good properties like tighter bound and smaller variance. The design choices not only result in a better performance of VHRNN models than VRNN but also enables VHRNNs to perform change detection at inference time and generalize to unseen dynamics. In contrast, VRNN models without dynamically generated weights failed on these tasks. The well-motivated composition of existing modules that results in better performance is a scientific contribution.
>
> Question:
> While the FIBO is mainly used as the objective in this paper, is the ELBO enough if the authors care the simultaneous low reconstruction error and low KL divergence?
> Answer:
> We use FIVO instead of ELBO for training and evaluating our models due to the advantages of FIVO over ELBO and fair comparison with previous works. The objective of our model is to maximize a variational lower bound of the data log-likelihood. FIVO is a tighter variational lower bound than ELBO with a smaller variance based on a well-studied particle filtering algorithm. Previous work [1] using VRNN trained and evaluated by FIVO demonstrates the state-of-the-art results for modeling time-series with latent variable models on various settings.
>
> Question: It is unclear and explained little if the comparison using parameter count is fair for VHRNN and VRNN since they have different structures.
> Answer:
> We acknowledge that it is difficult to define and compare the complexities of VHRNN and VRNN models because of the structural difference between them. However, the number of parameters is a widely adopted surrogate of model complexity of deep neural network and often used in the study of models’ parameter-performance efficiency[2]. We also show an additional plot of FIVO vs the number of hidden units in the current updated version in the appendix. We can see that VHRNN also dominates the performance of VRNN with a similar or fewer number of hidden units in most of the settings. Furthermore, the fact that VHRNN almost always outperforms VRNN for all parameter or hidden unit sizes precisely shows the superiority of the new architecture.
>
> Question: It would be nicer to discuss for which kind of time-series VRNN is enough.
> Answer:
> The experimental results on HT sensor dataset show that the VRNN model achieves comparable performance as VHRNN with a similar number of parameters. In contrast, VHRNN consistently outperforms VRNN on JSB Chorale dataset. It’s worth noting that in comparison with JSB Chorale dataset, HT Sensor has simpler data generation dynamics (there are only three types of stimulus applied to the sensors, white noise, alcohol, and banana) and the dataset size of HT sensor is much larger than JSB Chorale. So we believe VRNN models given enough capacity might perform as well as VHRNN on time series data with relatively simple data generation pattern and enough training data.However, for datasets with more complex patterns, VHRNN would be a more suitable model.
>
> We will update the response with the experimental results of VRNN with a latent dimension of 4 on the synthetic dataset as soon as possible and add a figure to illustrate equations in Section 3 in the upcoming version of our paper. Please reconsider the rating if we have satisfactorily answered the reviewer's questions. We are happy to answer any further questions the reviewer might have.
>
> [1] Maddison, Chris J., et al. "Filtering variational objectives." Advances in Neural Information Processing Systems. 2017.
> [2] Huang, Gao, et al. "Densely connected convolutional networks." Proceedings of the IEEE conference on computer vision and pattern recognition. 2017.

---

### Official Review · AnonReviewer1 · 2019-10-23
**Official Blind Review #1**

**Rating:** 6

**Review:**

This paper proposes a variational hyper recurrent neural network which is a combination of the variational RNN and the hypernetwork. The hypernetwork is an RNN whose output modifies the parameters of the variational RNN dynamically at runtime. Overall, this seems like an extension of the idea of using a hypernetwork with the VRNN (rather than the RNN as done in Ha. et. al). The model is trained via the FIVO objective. The model and learning algorithm are compared to the variational RNN and tested on a variety of synthetic settings where the VHRNN outperforms the VRNN in held-out likelihood. The performance gains are investigated on synthetic datasets where the paper notes that the VHRNN is often quicker to adapt variations that happen within seqences (for example, the paper considers a dataset where multiple patterns are stitched together into a sequence and study the changes in the KL divergence and reconstruction at switch points). On four real-world sequential datasets, the paper finds that the model outperforms the VRNN across many configurations and with a fewer number of parameters.

Summary: I don't think the model presented here is very novel, in that it is a combination of existing ideas; however, the paper does a good job of studying the model in a variety of different configurations on both synthetic and real-world data. The model does appear to consistently outperform the Variational RNN of Chung et. al.

Questions and comments:
(a) I do not think the word "System Identification" should be used on page 5 to describe the results of Figure 2. Doing so would overload existing notation in the time series literature where the word refers to the identification of parameters under a pre-specified physical system.
(b) How well does the Recurrent Hyper network (with no latent variable) do on the tasks considered here? I understand that it may be a less expressive model in general, but it is not clear to me why it would not be a competitive baseline on some of the smaller datasets considered here -- was this baseline tried?
(c) Did you experiment with non-temporal architectures for the hypernetwork? Since z_t and h_t-1 (which are conditioned on) contain information about the history of the sequence, one might argue that conditioning on them might suffice to predict the modifications to the parameters of the theta and g.


**Experience Assessment:**

I have published one or two papers in this area.

**Review Assessment: Checking Correctness Of Derivations And Theory:**

I assessed the sensibility of the derivations and theory.

**Review Assessment: Checking Correctness Of Experiments:**

I assessed the sensibility of the experiments.

**Review Assessment: Thoroughness In Paper Reading:**

I read the paper at least twice and used my best judgement in assessing the paper.

---

> ### Author Response · Authors · 2019-11-10
> **Response to AnonReviewer1**
>
> We would like to thank the reviewer for providing thoughtful comments. The reviewer recognized that our proposed models consistently outperform the state-of-the-art VRNN models with fewer parameters on both synthetic and real-world data. However, the motivation of our work is to better handle complex variabilities within and across sequences in time-series data by dynamically changing the model’s weights based on observations. Hypernetwork is the class of models of natural choice to handle dynamic weight generation. Our work introduces the idea of dynamic weight generation into the study of sequence modeling with variational Bayesian methods to achieve the goal. Our proposed models are trained with the state-of-the-art FIVO objective based on the well-studied particle filtering algorithm. Dynamic weight generation not only results in the better performance of VHRNN models then VRNN but enables VHRNNs to perform change detection at inference time and generalize to unseen dynamics. In contrast, VRNN models without dynamically generated weights failed on these tasks. The use of existing modules in our work is well motivated and it results in the advantage and superior performance of our model over VRNN.
>
> We will modify the wording  of ``System Identification’’ in the next updated version of our paper and update this response with more experiment results as soon as possible. We are also willing to answer any further questions the reviewer might have.

---

> > ### Author Response · Authors · 2019-11-15
> > **VHRNN-MLP vs VHRNN-RNN Parameter Performance Comparison**
> >
> >                 VHRNN-MLP vs VHRNN-RNN Parameter Performance Comparison
> > +-------------+----------+-----------------------------+-----------------------------+--------------------------+---------------------------+
> > | Datasets| Z Dim. | VHRNN-MLP Param. | VHRNN-RNN Param.| VHRNN-MLP FIVO | VHRNN-RNN FIVO |
> > +-------------+----------+-----------------------------+-----------------------------+--------------------------+---------------------------+
> > |                 | 12        | 28k                               | 27k                               | -6.83                        | -6.80                         |
> > +                 +----------+-----------------------------+-----------------------------+--------------------------+---------------------------+
> > |                 | 14        | 32k                               | 31k                               | -6.73                        | -6.76                         |
> > +                 +----------+-----------------------------+-----------------------------+--------------------------+---------------------------+
> > |                 | 16        | 37k                               | 35k                               | -6.76                        | -6.81                         |
> > +                 +----------+-----------------------------+-----------------------------+--------------------------+---------------------------+
> > | JSB          | 24        | 62k                               | 58k                               | -6.83                        | -6.86                          |
> > +                 +----------+-----------------------------+-----------------------------+--------------------------+---------------------------+
> > |                 | 32        | 94k                               | 88k                               | -7.13                        | -6.82                         |
> > +                 +----------+-----------------------------+-----------------------------+--------------------------+---------------------------+
> > |                 | 40        | 135k                             | 125k                            | -7.08                         | -6.87                         |
> > +                 +----------+-----------------------------+-----------------------------+--------------------------+---------------------------+
> > |                 | 42        | 146k                             | 135k                            | -7.02                         | -6.90                         |
> > +-------------+----------+-----------------------------+-----------------------------+--------------------------+---------------------------+
> > |                 | 12        | 16k                               | 16k                              | 9.47                          | 9.33                           |
> > +                 +----------+-----------------------------+-----------------------------+--------------------------+---------------------------+
> > |                 | 14        | 19k                               | 18k                              | 9.21                          | 9.34                           |
> > + Stock      +----------+-----------------------------+-----------------------------+--------------------------+---------------------------+
> > |                 | 16        | 24k                               | 23k                              | 9.31                          | 9.27                           |
> > +                 +----------+-----------------------------+-----------------------------+--------------------------+---------------------------+
> > |                 | 18         | 29k                              | 28k                              | 9.19                          | 9.31                           |
> > +-------------+----------+-----------------------------+-----------------------------+--------------------------+---------------------------+
> > | Syn-Test | 4          | 1676                            | 1568                             | -5.13                        | -4.68                          |
> > +-------------+----------+-----------------------------+-----------------------------+--------------------------+---------------------------+

---

> > ### Author Response · Authors · 2019-11-15
> > **VHRNN-MLP vs VHRNN-RNN Systematic Generalization Study Comparison**
> >
> >             VHRNN-MLP vs VHRNN-RNN Systematic Generalization Study Comparison
> > +------------------+----------+------------+-------+-----------------+-------------+----------+------------+------------------+---------+
> > | Model           | Z Dim. | Param. | Test  | NOISELESS | SWITCH | RAND  | LONG    | ZERO-SHOT | ADD   |
> > +------------------+----------+------------+-------+-----------------+-------------+----------+------------+------------------+---------+
> > | VHRNN-MLP| 4          | 1676      | -5.13 | -2.73            | -5.22      | -4.12    | -604937 | -2.94             | -3.84  |
> > +------------------+----------+------------+-------+-----------------+-------------+----------+------------+------------------+---------+
> > | VHRNN-RNN| 4         | 1568      | -4.68 | -2.08            | -4.27       | -3.91    | -3005     | -2.57             | -2.62  |
> > +------------------+----------+------------+-------+-----------------+-------------+----------+------------+------------------+---------+

---

> > ### Author Response · Authors · 2019-11-15
> > **VHRNN and HyperLSTM Comparison**
> >
> > VHRNN vs HyperLSTM Comparison with Same Number of Hidden Units
> > +--------------+-------------------+-------------------+--------------+
> > | Datasets  | Hidden Units| VHRNN FIVO | HLSTM LL|
> > +--------------+-------------------+-------------------+--------------+
> > |                   | 24                   | -6.80               | -9.15         |
> > +                  +-------------------+-------------------+--------------+
> > |                   | 28                   | -6.76                | -8.91        |
> > +                  +-------------------+-------------------+--------------+
> > |                   | 32                   | -6.81               | -8.87         |
> > +                  +-------------------+-------------------+--------------+
> > | JSB            | 48                   | -6.86                | -8.94         |
> > +                  +-------------------+-------------------+--------------+
> > |                  | 64                    | -6.82               | -8.85         |
> > +                  +-------------------+-------------------+--------------+
> > |                  | 80                    | -6.87               | -8.77         |
> > +                  +-------------------+-------------------+--------------+
> > |                  | 84                    | -6.90               | -8.77         |
> > +--------------+-------------------+-------------------+--------------+
> > |                  | 24                    | 9.33                | 8.61           |
> > +                  +-------------------+-------------------+--------------+
> > |                  | 28                    | 9.34                | 8.44           |
> > + Stock       +-------------------+-------------------+--------------+
> > |                  | 32                    | 9.27                | 8.44           |
> > +                  +-------------------+-------------------+--------------+
> > |                  | 36                    | 9.31                | 8.47           |
> > +--------------+-------------------+-------------------+--------------+
> >
> >                   VHRNN vs HyperLSTM Comparison with Similar Number of Parameters
> > +--------------+--------------------+----------------------+----------------------+-----------------------+-------------------+------------+
> > | Datasets | HLSTM Param.| HLSTM Hidden| VHRNN Param. | VHRNN Hidden | VHRNN FIVO | HLSTM LL|
> > +--------------+--------------------+----------------------+----------------------+-----------------------+-------------------+------------+
> > |                  | 29k                    | 48                       | 27k                     | 24                         | -6.80                | -8.94      |
> > +                  +--------------------+----------------------+----------------------+-----------------------+-------------------+------------+
> > | JSB           | 82k                    | 84                        | 88k                     | 64                        | -6.81                | -8.77      |
> > +                  +--------------------+----------------------+----------------------+-----------------------+-------------------+------------+
> > |                  | 124k                  | 104                     | 125k                   | 88                         | -6.87               | -8.85      |
> > +--------------+--------------------+----------------------+----------------------+-----------------------+-------------------+------------+
> > |                  | 15k                    | 36                       | 16k                      | 24                         | 9.33                | 8.47       |
> > + Stock       +--------------------+----------------------+----------------------+-----------------------+-------------------+------------+
> > |                  | 26k                    | 48                       | 23k                      | 32                         | 9.27                | 8.51       |
> > +--------------+--------------------+----------------------+----------------------+-----------------------+-------------------+------------+

---

> > ### Author Response · Authors · 2019-11-15
> > **Response to AnonReviewer1 Update**
> >
> > We thank the reviewer for making suggestion on the use of specific wording to avoid confusion. In the updated version of the paper, we used the term “regime identification” to refer to the model’s ability to uncover the underlying patterns of data generation.
> >
> > We ran experiments with VHRNN models on JSB Chorale, Stock and the synthetic dataset after replacing the recurrent network with an MLP for $\theta$ to generate the weights of $g$ and keeping the other components of VHRNN unchanged. We name this variant of VHRNN model VHRNN-MLP and the original VHRNN model VHRNN-RNN. We present the experiment results and comparisons with VHRNN-RNN models with the same latent dimension in a separate responses named "VHRNN-MLP vs VHRNN-RNN Parameter Performance Comparison " due to space constraint. We also present systematic generalization study results of VHRNN-MLP in various scenarios of the synthetic dataset in the response named "VHRNN-MLP vs VHRNN-RNN Systematic Generalization Study Comparison". The experiment results are also incorporated into Section I of the appendix in the updated version of our paper.
> >
> > As we can see, given the same latent dimension, VHRNN-MLP models can have slightly better performance than VHRNN-RNN with more parameters in some cases, VHRNN-MLP performs worse than VHRNN-RNN in more settings. However, the performance of VHRNN-MLP also degrades faster than VHRNN-RNN on the JSB Chorale dataset as the latent dimension increases. Systematic generalization study on the synthetic dataset also shows that VHRNN-MLP has worse performance than VHRNN-RNN no matter in Test setting or in the systematically varied settings.
> >
> > We ran experiments using our implementation of HyperLSTM proposed in the original HyperNetworks[1] on JSB Choral and Stock datasets. Compared with VHRNN, HyperLSTM does not have latent variables. It does not have an encoder or decoder either. Our implementation of HyperLSTM is similar to the recurrence model of VHRNN, defined in Equation 1, using LSTM cell. The output distribution is binary for JSB Chorale dataset and a mixture of Gaussian with 5 components for Stock dataset. Experiment results and comparisons against VHRNN are presented in a separate response named “VHRNN and HyperLSTM Comparison”. Since HyperLSTM does not have a latent variable, we compare VHRNN and HyperLSTM in terms of the number of hidden units and the total number of parameters. The number of hidden units in VHRNN and HyperLSTM models is twice the dimension of hidden state as both model contains two RNNs, one primary network and one hyper network. For HyperLSTM models, we report the exact log likelihood and for VHRNN models, we report the FIVO value which is a lower bound of log likelihood. As we can see, the performance of VHRNN completely dominates HyperLSTM no matter what number of hidden units we pick. Actually, the performance of HyperLSTM is even worse than VRNN which indicates the importance of latent variable in modelling complex time-series data. More implementation details and experiment results on HyperLSTM models can be found in Section H of the appendix in the updated version of the paper.
> >
> > We will add experiment results of VHRNN-MLP and HyperLSTM models on more datasets and comprehensive comparisons with VHRNN-RNN in future versions. We hope our response answers the reviewer’s questions and satisfies the reviewer’s requests.
> >
> > [1]Ha, David, Andrew Dai, and Quoc V. Le. "Hypernetworks." arXiv preprint arXiv:1609.09106 (2016).

---

### Official Review · AnonReviewer4 · 2019-10-29
**Official Blind Review #4**

**Rating:** 6

**Review:**

In this paper the authors propose an architecture based on variational autoencoders and hyper-networks. The basic idea is that the weights of the underlying RNN/autoencoder are not fixed, but are coming from another RNN/feed-forward network which captures the underlying dynamics and adjusts the weights accordingly. The experimental results show the benefit of the model compared to a similar method without hypernets.

In terms of novelty, the combination of auto-encoder RNNs and hyper-networks is not entirely novel and it has previously been developed (https://www.biorxiv.org/content/10.1101/658252v1). However, while I think these previous works should be discussed in the paper (they are not currently), the two architectures are sufficiently different and the current work is novel enough in my opinion. On the other hand, in terms of presentation, I think the paper can be improved. The architecture is not entirely clear from the text. I think a graph showing the architecture of the model would be very helpful here. The notations also seem loosely defined (what is the dimensionality of x_t, z_t, etc.) and sometimes undefined (e.g., x_t in equation 1 is not defined).

In terms of model architecture, it wasn’t clear for me why the hyper-network for \phi is feedforward but the one for \theta is RNN?

The experiments seem promising, but I have the following questions before being able to assess the results:

- How many LSTM units are in the model in each experiment? Are they similar in both VHRNN and VRNN?

- What is the structure of encoder/decoder layers? How many units in each layer? Are they the same for VHRNN and VRNN?

-There are four sets of weights in the primary model: weights of RNN, dec, enc, prior. How are these weights generated by the two hyper-networks \theta and w?

- How is the number of parameters in the experiment calculated. Do they refer to the number of parameters in the hypernetworks only?


Minor:
I suspect the spaces between the equations and captions are also manually changed which has made the paper physically dense and a bit unreadable (see equation 4 for example). Similarly, the space between the caption of Figure 1 and the text after seems too small.




**Experience Assessment:**

I have published one or two papers in this area.

**Review Assessment: Checking Correctness Of Derivations And Theory:**

I did not assess the derivations or theory.

**Review Assessment: Checking Correctness Of Experiments:**

I assessed the sensibility of the experiments.

**Review Assessment: Thoroughness In Paper Reading:**

I made a quick assessment of this paper.

---

> ### Author Response · Authors · 2019-11-10
> **Response to AnonReviewer4**
>
> We would like to thank the reviewer for recognizing the contributions of our work and making suggestions on the presentation of the paper. We completely agree that adding a figure showing the architecture of hyper LSTM and better clarifying the definition of notations would be helpful to the readers. They will be incorporated into the next updated version of our paper soon. We will also improve the presentation, cite existing related works and add more experiment details in the next version. We are willing to answer any further questions the reviewer might have.
>
> Question: Why the hyper-network for \phi is feedforward but the one for \theta is RNN?
> Answer:
> As the generation of both samples and network weights is determined by the latent variable and hidden states, we assume that they encode information about the current underlying dynamics or patterns of data generation. Therefore, we can use the latest latent variable and hidden state with a feed-forward network to determine the weights of the decoder $\phi$ that generates the observation for one step. Meanwhile, we would like to explicitly keeping track of the history of data generation dynamics for the hyper network. We believe it could better help the hyper network to produce weights that can adapt to the change between the old dynamic and the new one. Moreover, the design of hyper networks that uses an RNN to generate the weights of another RNN has been well studied in the original HyperNetworks and demonstrates promising performance on many tasks. Thus, we use another RNN as the hyper network to generate the weights $\theta$ of the primary RNN. It takes the latent variable and the previous hidden state from the primary RNN as input.
>
> Question: How many LSTM units are in the model in each experiments? Are they similar in both VHRNN and VRNN?
> Answer:
> The hidden state dimension of each LSTM network in the paper is the same as the latent variable dimension except for the ablation study section. However, it’s worth noting that VHRNN contains two LSTM networks, the primary network and the hyper network.  They have the same number of hidden units unless otherwise mentioned. Therefore, the number of hidden units in a VHRNN model is twice the number of latent dimension. We used one-layer LSTM for both VHRNN and VRNN models. In the current updated version of our paper, we also show plots of FIVO vs number of hidden units for VRNN and VHRNN models in Section E of the appendix. We can see that VHRNN also dominates the performance of VRNN with a similar or fewer number of hidden units in most of the settings.
>
> Question: What is the structure of encoder/decoder layers? How many units in each layer? Are they the same for VHRNN and VRNN?
> Answer:
> We follow similar principles as the original Variational RNN paper when designing the encoder and decoder. The architecture of the encoder defined in Equation 5 is the same for VHRNN and VRNN. For synthetic datasets, we used a fully connected network with two hidden layers; each has the same number of units as the latent variable dimension. For other real-world datasets, we use a fully connected network with one hidden layer with the same number of units as the latent dimension.
> For VHRNN models, we use fully connected hyper networks with two hidden layers for synthetic data and fully-connected hyper networks with one hidden layer for other datasets as the decoder networks. The number of units in each hidden layer is also the same as the latent variable. For VRNN models, we use plain feed-forward networks for decoder. The number of hidden layers and units in the hidden layer are determined in the same way as VHRNN models.
> We will add the experiment details for different settings in the next version of our paper.
>
> Question: There are four sets of weights in the primary model: weights of RNN, dec, enc, prior. How are these weights generated by the two hyper-networks \theta and w?
> Answer:
> The weights of the RNN and encoder are dynamically generated while the weights of the encoder and prior networks are fixed. $\omega$ denotes the weight of the decoder. For each layer of the decoder, we generate the bias and a scaling vector that scales each row of the weight matrix using a two-layer MLP (hyper network). $\theta$ denotes the weight of the RNN part of our model. At each time step, we generate the bias vectors and row-scaling vectors for all weight matrices of the primary RNN by directly mapping from the hidden state of the hyper RNN via linear transformations.
>
> Question: How is the number of parameters in the experiment calculated. Do they refer to the number of parameters in the hypernetworks only?
> Answer:
> The number of parameters reflects the total number of trainable parameters of the model, including the RNN, prior network, the encoder which proposes the variational posterior distribution and the decoder. For VHRNN, the parameter counts of the RNN and the decoder include both the primary network and the hyper network.

---

> > ### Author Response · Authors · 2019-11-15
> > **Response to AnonReviewer4 Update**
> >
> > We thank the reviewer for making suggestion of a related work and recognizing our work’s difference from that one and unique contribution. The latest related work has been added to the Background and Related Work section with a brief discussion on the similarity and difference between their work and ours. We further improved the Model Formulation section by clarifying the definition of all the notations. We showed one implementation of VHRNN’s recurrence model $g$ using LSTM cell in Section 3 in the original submission. We visualize the architecture of this implementation in Section A of the appendix in the updated version of our paper. We hope the diagram could further illustrate this implementation of VHRNN. The readability of the paper is improved after the update with more space between equation, caption and text. We will continue improving the paper’s presentation in future versions.
> >
> > As AnnoReviewer1 suggested, we also ran experiments where the hypernetwork for $\theta$ is replaced by a feed-forward network without temporal structure. The results are presented in the response to AnnoReviewer1. We find that VHRNN models with a feed-forward network for $\theta$ overall performs slightly worse than VHRNNs with recurrent network for $\theta$ given the same latent dimension and similar number of parameters. However, in systematic generalization study using the synthetic dataset, we find VHRNN models with a feed-forward network for $\theta$ uniformly shows worse performance than the VHRNN models with a recurrent network structure in all scenarios. These experiment results further justifies the use of a recurrent network for $\theta$.
> >
> > We hope our response answers the reviewer’s questions and satisfies the reviewer’s requests.

---

### Author Response · Authors · 2019-11-15
**Summary of Changes to the Paper**

We thank all the reviewers for their thoughtful comments. Upon the suggestion from the reviewers, we made the following major changes to the content of our paper:

1. We further clarified the definition of notations in Section 3 Model Formulation.

2.  We replaced the term “system identification” with “regime identification” that refers to the model’s ability to uncover the underlying dynamics of data generation.

3. We added the systematic generalization study results of VRNN models with latent dimension of 4 to Tab. 1

4. We added a diagram of VHRNN’s recurrence model using LSTM cell to better illustrate the implementation of VHRNN in Section A of the appendix.

5. We made performance vs number of hidden units comparisons between VHRNN and VRNN models and added the plots showing the comparison results in Section F of Appendix.

6. We provided more implementation details about the encoder, decoder and prior networks of VHRNN as well as the observation and latent variable encoding networks in the appendix.

7. We compared our VHRNN against HyperLSTM on two real-world datasets and added the comparison results to Section H of the sppendix.

8. We compared our proposed VHRNN model with an variant of VHRNN that has no recurrence structure in the hyper networks. The comparison results and analysis are presented in Section I of the appendix.

We hope our responses and updated paper satisfactorily addressed the reviewers’ questions and concerns.

---

### Decision · Program_Chairs · 2019-12-19

**Decision:**

Reject

**Comment:**

The paper proposes a neural network architecture that uses a hypernetwork (RNN or feedforward) to generate weights for a network (variational RNN), that models sequential data. An empirical comparison of a large number of configurations on synthetic and real world data show the promise of this method.

The authors have been very responsive during the discussion period, and generated many new results to address some reviewer concerns. Apart from one reviewer, the others did not engage in further discussion in response to the authors updating their paper.

The paper provides a tweak to the hypernetwork idea for modeling sequential data. There are many strong submissions at ICLR this year on RNNs, and the submission in its current state unfortunately does not pass the threshold.